# Type-I Hemins and Free Porphyrins from a Western Australian Sponge *Isabela* sp.

**DOI:** 10.3390/md21010041

**Published:** 2023-01-03

**Authors:** Samuele Sala, Stephen A. Moggach, Gareth L. Nealon, Jane Fromont, Oliver Gomez, Daniel Vuong, Ernest Lacey, Gavin R. Flematti

**Affiliations:** 1School of Molecular Sciences, The University of Western Australia, Crawley, WA 6009, Australia; 2Australian National Phenome Centre and Centre for Computational and Systems Medicine, Health Futures Institute, Murdoch University, Harry Perkins Building, Perth, WA 6150, Australia; 3Centre for Microscopy, Characterisation and Analysis, The University of Western Australia, Crawley, WA 6009, Australia; 4Collection and Research, Western Australian Museum, Welshpool, WA 6106, Australia; 5Microbial Screening Technologies Pty. Ltd., Smithfield, NSW 2164, Australia

**Keywords:** porphyrin, *Isabela*, hemin, X-ray diffraction, NOESY NMR, cytotoxic

## Abstract

Two novel free porphyrins, isabellins A and B, as well as the known compounds corallistin D and deuteroporphyrin IX were isolated from a marine sponge *Isabela* sp. LC-MS analysis of the crude extract revealed that the natural products were present both as free porphyrins and iron(III) coordinated hemins, designated isabellihemin A, isabellihemin B, corallistihemin D and deuterohemin IX, respectively. Structures were determined via high-resolution mass spectrometry, UV-Vis spectroscopy and extensive NOESY NMR spectroscopic experiments. The type-I alkyl substitution pattern of isabellin A and isabellihemin A was assigned unambiguously by single crystal X-ray diffraction. Biological evaluation of the metabolites revealed potent cytotoxicity for isabellin A against the NS-1 murine myeloma cell line.

## 1. Introduction

Tetractinellid sponges of the *Corallistes* genus (Order: Tetractinellida; Family: Corallistes) have been reported to yield the microtubule stabilising macrolactone dictyostatin [1], and the poly-nitrogen compound corallistine [2], as well as the free porphyrins corallistins A-E and deuteroporphirin IX [3,4]. The assigned structures of corallistins A, B, C and E have been confirmed via total synthesis [5,6]. Recently, taxonomic re-identification of the sponges reported to produce corallistins A-E has suggested that the sponges are in fact members of the genus *Isabela* [7]. 

Metallated porphyrins and related macrocycles are expressed in most living organisms while synthetic porphyrins have been applied broadly in the area of photodynamic therapy [8]. They are however rarely encountered as functionalised secondary metabolites constituting a significant proportion of an organism’s metabolic extract [4]. Given the biological importance of porphyrins as by-products of heme biosynthesis, their biosynthetic pathway has been well studied [9,10]. Beginning with glycine and succinyl-CoA, the enzymes ALA-synthase and ALA-dehydratase yield porphobilinogen in animals, fungi and α-proteobacteria. In plants, Archea and most other Bacteria, porphobilinogen is biosynthesised from two tRNA bound glutamyl starter units [10]. The enzyme PBG-deaminase then leads to the linear hydroxymethylbilane. From here, cyclisation can either occur chemically to afford the D_4h_ symmetrical uroporphyrinogen-I, or enzymatically with D-ring inversion via the UPG-III synthase, PBG-deaminase complex to afford uroporphirinogen-III [9,11]. In the case of uroporphirinogen-III, stepwise decarboxylation via the UPG-III decarboxylase enzyme affords coproporphyrinogen-III [11], which then loses a further two CO_2_ groups via the CPG-oxidase enzyme leading to the formation of protoporphyrinogen IX [11]. From here, PPG-oxidase removes six hydrogens to give protoporphyrin IX, whereby two protons are substituted for one unit of Fe^2+^ via a ferrochelatase enzyme to afford heme B [10,11]. 

Our ongoing investigations [12,13] into the marine sponges of the Western Australian Marine Bio-resources Library (WAMBL) [14] yielded two novel free porphyrins from an *Isabela* marine sponge, herein named isabellins A and B (**1a**, **2a**, Figure 1). LC-MS analysis of the crude extract revealed that the compounds were present as both free porphyrins and as ferric hemin compounds (**1b, 2b**). Further analysis of the mixture revealed the presence of the known compounds corralistin D (**3a**) and deuteroporphyrin IX (**4a**) [4] as well as their corresponding hemin counterparts (**3b, 4b**). Nuclear magnetic resonance experiments of the compounds were complicated by intense signal suppression of the nuclei present on the aromatic scaffolds, as well as the difficulty in replication of experiments, presumably due to the effects of aromatic ring current on the rapidly aggregating/de-aggregating porphyrins in solution, as has been noted in prior literature [3,4,5]. To this end, the structures of the compounds were assigned unambiguously via detailed interpretation of 2D-NOESY experiments and single crystal X-ray crystallography of **1a/b**. The paramagnetic iron(III) hemin complexes **2b**, **3b** and **4b** were provisionally assigned via HR-ESI-MS. Biological evaluation of the isolated metabolites revealed compound **1a** as a potent inhibitor of the NS-1 murine myeloma cell line with strong selectivity for mammalian cell lines over bacterial pathogens and other Eukaryotes.

## 2. Results

HPLC-photo diode array detector analysis of the crude solvent extract of the Isabela sponge revealed a series of etio-type porphyrins with a characteristic Soret band at approximately 390 nm and Q-bands with intensities in the order of IV > III > II > I [15]. Large scale isolation of the compounds was achieved using a combination of normal phase chromatography and reversed phase HPLC under acidic conditions.

Compound **1a** was isolated as a dark red crystalline solid. HR-ESIMS gave a protonated molecular ion [M + H]^+^ at *m*/*z* 367.1924 consistent with the molecular formula C_24_H_23_N_4_ (requires 367.1923). To our surprise, ^1^H NMR of the compound in CDCl_3_ (600 MHz) revealed the presence of a compound with four-fold rotational symmetry as revealed by three unique electronic environments (δ_H_ 10.12; CH-5, 9.14; CH-3, 3.76; C-2-CH_3_). We note that appropriate ^1^H NMR integration was achieved by applying a 10 second relaxation time to the pulse sequence. All attempts to obtain reliable ^13^C NMR data for compound **1a** were hampered by significant signal suppression, presumably caused by the significant aromatic ring current present in the system. The difficulty in obtaining reliable ^13^C NMR spectra for free base porphyrins has been documented elsewhere [5]. The structure of compound **1a** was subsequently assigned unambiguously via single crystal X-ray diffraction (Figure 2) and assigned the trivial name isabellin A. The extremely reduced state of compound **1a** is a biosynthetically anomaly, and to the best of our knowledge is the first report of a highly reduced geo-porphyrinoid [16] isolated from a living organism.

Compound **1b** was isolated as a brown solid which recrystallised in chloroform to give dark brown needles. HR-ESIMS of the compound gave a molecular ion cluster [M − 2H + Fe(III)]^+^ with an isotopic distribution characteristic of an iron atom at *m*/*z* 420.1032, consistent with the molecular formula C_24_H_20_N_4_^56^Fe^+^ (requires 420.1037), as well as a prominent acetonitrile adduct at *m*/*z* 461.1302 consistent with the molecular formula C_26_H_23_N_5_^56^Fe^+^ (requires 461.1303), allowing us to conclude that we had isolated the ferric hemin counterpart to compound **1a**. The ^1^H and ^13^C NMR analysis of the paramagnetic Fe(III) complex proved unproductive. Ultimately, the structure of compound **1b** was assigned via single crystal X-ray diffraction as the trifluoroacetate (TFA) salt (Figure 3).

Compound **2a** was isolated as dark brown amorphous solid. HR-ESIMS gave a protonated molecular ion [M + H]^+^ at *m*/*z* 439.2133 consistent with the molecular formula C_27_H_27_N_4_O_2_ (requires 439.2134). The ^1^H NMR analysis of the compound in acidified DMSO-*d*_6_ revealed the presence of four meso-proton environments (δ_H_ 10.37–10.34), three pyrrolic protons (δ_H_ 9.39–9.34), four methyl groups (δ_H_ 3.77, 3.75, 3.73 and 3.66) and a single propionate group as evidenced by two broad spin coupled triplets at δ_H_ 4.40 and δ_H_ 3.21. As with compound **1a**, extensive ^13^C NMR experiments failed to resolve the majority of the ^13^C nuclei attributable to a compound of this size. A range of NMR solvents and experimental parameters were trialled.

Given the capacity of the organism in question to produce porphyrins with both type-I (derived from a symmetrical APAPAPAP bearing uroporphirinogen precursor, with A = Acetyl and P = Propionyl) and type-III (derived from an asymmetrical APAPAPPA bearing uroporphirinogen precursor, featuring D-ring inversion) topology, configurational assignment of the alkyl substituents present on compound **2a** became a non-trivial exercise, with one potential type-I isomer and four potential type-III isomers possible. After some consideration it became clear that a simple experiment would be able to distinguish between naturally occurring type-I and type-III configurational isomers with four methyl groups present. The method can be summarised with the conditional statement: If every meso-proton of the porphyrin macrocycle shows a 2D-NOE correlation to a corresponding methyl group, then the porphyrin must be a type-I derived porphyrin. Extending the rationale, if one meso-proton does not show a correlation to a methyl group, then the porphyrin is either a natural type-III derived porphyrin, or a type-IV porphyrin, of which there are no naturally occurring derivatives.

Porphyrin **2a** showed clear correlations from each meso-proton to each methyl group, and was therefore assigned the type-I structure depicted (Figure 4). The compound was given the trivial name isabellin B (The authors here suggest that the new trivial name Isabellin be used to designate type-I porphyrins within the lithistid family of compounds, whereas the trivial name Corallistin be retained for porphirins with a type-III alkyl substitution pattern.). In similar fashion to **2a**, the type-III alkyl substitution patterns of the known metabolites corallistin D (**3a**) and deuteroporphyrin IX (**4a**) were confirmed using the methodology described above.

Reanalysis of the sponge crude extract by LC-HRMS revealed a series of ferric metabolites consistent in accurate mass measurements to be the hemin counterparts of **2a**, **3a** and **4a**, in similar relationship to that between **1a** and **1b** (Table 1). Given our previous difficulty in obtaining ^1^H and ^13^C NMR data for the paramagnetic **1b**, isolation of the compounds was not pursued. In support of their identity as the ferric counterparts to **2a**, **3a** and **4a,** treatment of the crude extract with concentrated H_2_SO_4_ led to the disappearance of the ferric metabolites when analysed by LC-MS. No new peaks (aside from those ascribed to **2a**, **3a** and **4a**) were seen in the chromatogram allowing for the provisional assignment of **2b**, **3b** and **4b** as illustrated.

Biosynthetically, compounds **1a/b** and **2a/b** are presumably derived from uroporphyrinogen-I, whereas compounds **3a/b** and **4a/b** are divergent uroporphyrinogen-III derivatives. Given the degree of elaboration observed on the respective type-I and type-III derived compounds, it seems that **1a/b**, **2a/b** act as more efficient substrates for the UPG-decarboxylase, CPG-oxidase and PPG oxidase enzymes, as well as the subsequent reducing and devinylating enzymes active on the scaffolds. Whether the iron was chelated in its ferric state as isolated or in the ferrous state as has been reported for ferrochelatase enzymes [10,11] remains unknown, however LC-MS analysis of the sponge crude extract failed to detect any iron(II) metabolites.

Testing of the metabolites against a panel of micro-organisms and cell lines (Table 2 and Table 3) revealed potent cytotoxicity of compound **1a** against the NS-1 myeloma cell line, with an MIC of 0.4 μg/mL at the 72-h time-point and 0.8 μg/mL at the 96-h time-point, comparable to the sparsomycin positive control. Testing against the neonatal foreskin fibroblast (NFF) cell line revealed marginal selectivity for the tumorigenic cell line with an MIC of 1.6 μg/mL at both time-points tested. In addition to this the compound was found to be moderately bacteriostatic towards the Gram-positive pathogens *Bacillus subtilis* and Staphylococcus aureus with an MIC of 25 μg/mL at the 24-h time-point, and inhibited the growth of *Giardia duodenalis* (MIC = 6.3 μg/mL). In stark contrast, metabolites **1b**, **2a**, **3a** and **4a** failed to display significant activity against any of the organisms and cell-lines tested, with the exception of mild activity towards NS-1 displayed by compound **3a** (MIC = 50 μg/mL at the 72-h time-point), and mild anti-bacterial activity displayed by compound **4a** towards *Staphylococcus aureus* (MIC = 50 μg/mL at the 72-h time-point). The cytotoxicity displayed by compound **1a** is in line with other porphyrin macrocycles [8]. We postulate that the discrepancy in activity between metabolite **1a** and other metabolites tested may be in part due to an increase in membrane permeability afforded to **1a** by its lipophilic structure, over that of the Fe(III) salt **1b** and the carboxylate bearing **2a**, **3a** and **4a**.

## 3. Discussion

Two new free porphyrins, isabellins A (**1a**) and B (**2a**), an iron (III) coordinated porphyrin, isabellihemin A (**1b**), and the known compounds corallistin D (**3a**) and deuteroporphyrin IX (**4a**) were isolated from a marine sponge *Isabela* sp. The type-I alkyl substitution pattern of **1a**, **1b** and **2a** was assigned unambiguously by NOESY NMR spectroscopic experiments and single crystal X-ray diffraction (Appendix A). Testing of the isolated metabolites against a panel of micro-organisms and cell lines revealed potent cytotoxicity for **1a** against the NS-1 and NFF cell lines that was comparable to that of the sparsomycin positive control.

## 4. Materials and Methods

### 4.1. General Experimental

UV/Vis spectra were acquired on an Agilent Cary 60 UV/Vis spectrometer. Nuclear magnetic resonance (NMR) spectra were recorded on a Bruker Avance IIIHD 500 MHz spectrometer (500.1 MHz for ^1^H and 125.8 MHz for ^13^C) and a Bruker Avance IIIHD 600 MHz spectrometer (600.1 MHz for ^1^H and 150 MHz for ^13^C). Chemical shifts were calibrated against the residual solvent present: in CDCl_3_ (^1^H, δ 7.26 and ^13^C, δ 77.16 ppm), in CD_3_OD (^1^H, δ 3.31 and ^13^C, δ 49.00 ppm), in (CD_3_)_2_SO (^1^H, δ 3.50 and ^13^C, δ 39.52 ppm), in (CD_3_)_2_CO (^1^H, δ 2.05 and ^13^C, δ 29.84 ppm) and expressed relative to TMS [17]. NMR spectra measured in neat TFA were measured at 273 K. Deuterium lock was maintained via insertion of a (CD_3_)_2_CO standard capillary tube insert. H_2_O was suppressed via selective presaturation at 11.5 ppm. HPLC-mass spectrometry and HRMS were conducted using a Waters Alliance e2695 HPLC connected to a Waters 2998 diode array detector and Waters LCT Premier XE time-of-flight mass spectrometer using either an atmospheric pressure chemical ionization (APCI) source or an electrospray ionisation (ESI) source in either positive or negative mode. HRMS was conducted with either APCI or ESI in W-mode, using leucine enkephalin (200 pg/μL) as internal lock mass. For LC-MS separation an Altima C_18_ column (150 mm × 2.1 mm, 5 μm, Grace Discovery Sciences, Columbia, MD, USA) was used with a flow rate of 0.3 mL/min. Rapid silica filtration (RSF) under reduced pressure was conducted on a sintered glass column using chromatographic silica (Davisil LC60A 40-63 micron, Grace Discovery Sciences, Columbia, MD, USA). Flash silica chromatography was conducted using the Reveleris X2 flash chromatography system equipped with a cartridge containing silica gel as the stationary phase (120 g, 40 μm, p/n 145). Semi-preparative and analytical HPLC were performed using either an Agilent 1200 HPLC system with a diode array detector (DAD) and fraction collector or using a Hewlett Packard 1050 equipped with a DAD and Pharmacia Biotech RediFrac fraction collector. Analytical work was conducted using an Apollo C_18_ reversed phase column (250 mm × 4.6 mm, 5 μm, Grace Discovery Sciences) utilising 20 μL injections, and with a flow rate of 1.0 mL/min, and semi-preparative HPLC was undertaken with an Apollo C_18_ reversed-phase column (250 mm × 10 mm, 5 μm, Grace Discovery Sciences) with 300 μL injections at a flow rate of 4.0 mL/min.

### 4.2. Characterisation

A specimen of *Isabela* sp. (WAM Z35787) was collected at 97 m depth on hard substrate off Zuytdorp, WA (27°03′06″ S, 113°06′03″ E) by Sherman sled on 5 December 2005 aboard the CSIRO research vessel Southern Surveyor, and was stored frozen at −18 °C at the Western Australian Museum. The ethanol preserved portion of the sponge is a thick, erect plate 900 mm tall, 500 mm wide and 300 mm thick. It has a stony, incompressible texture and a smooth surface; it was dark purple alive and black-brown in ethanol, and it stains the ethanol dark brown. The spicules are desmas, blunt-ended oxeas 115 × 5 μm, blunt-ended dichotriaenes 1130 μm long, microrhabds 40 μm long and thin spirasters 20 μm long.

### 4.3. Extraction and Isolation

A portion of the *Isabela* sp. sponge (14.9 g, frozen weight) was sheared with scissors and extracted overnight, three times in MeOH:DCM (1:1, *v*/*v*) solution (3 × 500 mL) to which was added TFA (3 × 0.5 mL). The crude dark brown extract was filtered (Whatman No. 1, 18.5 cm) and reduced *in vacuo* to give a dark brown gum. The sample was reconstituted in acidified CH_2_Cl_2_ (1.0% TFA) and absorbed on celite (1 g) before separating with a Reveleris automated flash chromatography module. The column was eluted with an isocratic solvent system consisting of 100% EtOAc for 20 min. The mobile phase was then increased from EtOAc to 100% MeOH over a further 5 min and held for 10 min. The flow rate was set at 25 mL/min and fractions were collected in 25 mL aliquots throughout the run. The fraction eluting at six minutes was separated using semi-preparative reversed phase HPLC, eluting with an isocratic solvent system of 95% ACN/H_2_O with 0.1% TFA to yield **1a** (3.5 mg). The fraction eluting at seven minutes was separated using semi-preparative reversed phase HPLC at a flow rate of 4 mL/min, eluting with an isocratic solvent system of 75% ACN/H_2_O with 0.1% TFA over 40 min to yield **1b** (2.2 mg) eluting at 5 min and **2a** (3.2 mg) eluting between 15–20 min. The fraction eluting at twelve minutes was separated using semi-preparative reversed phase HPLC at a flow rate of 4 mL/min, eluting with an isocratic solvent system of 55% ACN/H_2_O with 0.1% TFA over 40 min to yield **3a** (5.0 mg) and **4a** (4.2 mg).

### 4.4. Compound Characterization 

*Isabellin A***(1a)** dark red plate crystals, green in solution, pink when observed through a transmitted source of white light, bright red in acidified solution; UV/Vis (MeOH) λ_max_ (log ε) 390 (2.23), 495 (0.14), 530 (0.10), 565 (0.08), 615 (0.05) nm; ^1^H NMR (CDCl_3_, 600 MHz) δ [ppm] 10.12 (s, 1H), 9.14 (s, 1H), 3.76 (s, 3H); HRMS (ESI): *m/z* 367.1924 [M + H]^+^ (calcd for C_24_H_23_N_4_, 367.1923)*Isabellihemin A***(1b)** dark brown needles, brown in solution, UV/Vis (MeOH) λ_max_ (log ε) 385 (2.40), 490 (0.15), 605 (0.07) nm; HRMS (ESI): *m/z* 420.1032 [M − 2H + Fe(III)]^+^ (calcd for C_24_H_20_N_4_^56^Fe, 420.1037), *m/z* 461.1302 [M − 2H + ACN + Fe(III)]^+^ (calcd for C_26_H_23_N_5_^56^Fe, 461.1303)*Isabellin B***(2a)** Maroon solid, bright red in acidified solution; UV/Vis (MeOH) λ_max_ (log ε) 390 (0.74), 495 (0.06), 527 (0.04), 565 (0.03), 613 (0.02) nm; ^1^H NMR (DMSO-d_6_, 600 MHz) δ [ppm] 10.37 (s, 1H), 10.36 (s, 1H), 10.35 (s, 1H), 10.34 (s, 1H), 9.39 (s, 1H), 9.36 (s, 1H), 9.34 (s, 1H), 4.40 (bt, 2H), 3.77 (s, 3H), 3.76 (s, 3H), 3.73 (s, 3H), 3.66 (s, 3H), 3.20 (bt, 2H); HRMS (ESI): *m/z* 439.2133 [M + H]^+^ (calcd for C_27_H_27_N_4_O_2_, 439.2134)*Isabellihemin B***(2b)** HRMS (ESI): *m/z* 492.1248 [M − 2H + Fe(III)]^+^ (calcd for C_27_H_24_N_4_O_2_^56^Fe, 492.1249), *m/z* 533.1522 [M − 2H + ACN + Fe(III)]^+^ (calcd for C_29_H_27_N_5_O_2_^56^Fe, 533.1514)*Corallistin D***(3a)** Dark brown solid, bright pink in acidified solution: UV/Vis (MeOH) λ_max_ (log ε) 385 (3.55), 425 (0.21), 560 (0.26) nm; ^1^H NMR (CDCl_3_, 600 MHz) δ [ppm] 11.07 (bs, 1H) 10.69 (s, 1H), 10.66 (s, 1H), 10.65 (s, 1H) 9.40 (s, 1H), 4.46 (bt, 4H), 4.16 (q, 2H), 3.79 (s, 3H), 3.69 (s, 3H), 3.682 (s, 3H), 3.680 (s, 3H), 3.32 (bt, 4H), 2.41 (t, 3H); HRMS (ESI): *m*/*z* 539.2653 [M + H]^+^ (calcd for C_32_H_35_N_4_O_4_, 539.2658)*Corallistihemin D***(3b)** HRMS (ESI): *m/z* 592.1772 [M − 2H + Fe(III)]^+^ (calcd for C_32_H_32_N_4_O_4_^56^Fe, 592.1773), *m/z* 633.2034 [M − 2H + ACN + Fe(III)]^+^ (calcd for C_34_H_35_N_5_O_4_^56^Fe, 633.2038)*Deuteroporphyrin IX***(4a)** Dark brown solid, bright pink in acidified solution; UV/Vis (MeOH) λ_max_ (log ε) 390 (2.63), 500 (0.15), 525 (0.15), 560 (0.14), 600 (0.08) nm; ^1^H NMR (CDCl_3_, 600 MHz) δ [ppm] 11.10 (bs, 1H), 10.72 (s, 1H), 10.69 (s, 1H), 10.67 (s, 1H), 9.42 (s, 1H), 9.41 (s, 1H), 4.48 (bt, 4H), 3.80 (s, 3H), 3.79 (s, 3H), 3.71 (s, 3H), 3.68 (s, 3H), 3.22 (bt, 2H), 3.20 (bt, 2H); HRMS (ESI): *m/z* 511.2345 [M + H]^+^ (calcd for C_30_H_31_N_4_O_4_, 511.2345)*Deuterohemin IX***(4b)** HRMS (ESI): *m/z* 564.1474 [M − 2H + Fe(III)]^+^ (calcd for C_30_H_28_N_4_O_4_^56^Fe, 564.1460), *m/z* 605.1718 [M − 2H + ACN + Fe(III)]^+^ (calcd for C_32_H_31_N_5_O_4_^56^Fe, 605.1725) 

### 4.5. X-ray Crystallographic Analysis of ***1a***

From a solution of MeOH:DCM (3:1) blood red plates were obtained, A suitable crystal with dimensions 0.08 × 0.05 × 0.04 mm^3^ was selected and mounted on a XtaLAB Synergy, Single source at home/near, HyPix diffractometer. The crystal was kept at a steady *T* = 150.00(10) K during data collection. The structure was solved with the **ShelXT** 2018/2 [18] solution program using dual methods and by using **Olex2** 1.5 [19] as the graphical interface. The model was refined with **XL** [20] using full matrix least squares minimisation on ***F*^2^**.

C_24_H_22_N_4_, *M_r_* = 366.45, monoclinic, *P*2_1_/*c* (No. 14), a = 11.7999(5) Å, b = 10.7615(5) Å, c = 14.9068(6) Å, *b* = 96.745(4)°, *a* = *g* = 90°, *V* = 1879.83(14) Å^3^, *T* = 150.00(10) K, *Z* = 4, *Z′* = 1, *m*(Cu K*_a_*) = 0.609, 18158 reflections measured, 3906 unique (R_int_ = 0.0539) which were used in all calculations. The final *wR_2_* was 0.2031 (all data) and *R_1_* was 0.0621 (I ≥ 2 *s*(I)).

### 4.6. X-ray Crystallographic Analysis of 1b

From a solution of CHCl_3_ dark brown needles were obtained. A suitable crystal with dimensions 0.10 × 0.05 × 0.03 mm^3^ was selected and mounted on a XtaLAB Synergy, Single source at home/near, HyPix diffractometer. The crystal was kept at a steady *T* = 120.00(14) K during data collection. The structure was solved with the **ShelXT** 2018/2 [18] solution program using dual methods and by using **Olex2** 1.5 [19] as the graphical interface. The model was refined with **XL** [20] using full matrix least squares minimisation on ***F*^2^**.

C_25_H_20_F_1.5_FeN_4_O_1.5_, *M_r_* = 484.80, monoclinic, *P*2_1_/*c* (No. 14), a = 9.6505(10) Å, b = 15.664(2) Å, c = 17.1896(19) Å, *b* = 100.170(10)°, *a* = *g* = 90°, *V* = 2557.7(5) Å^3^, *T* = 120.00(14) K, *Z* = 4, *Z′* = 1, *m*(Cu K*_a_*) = 5.029, 13005 reflections measured, 1627 unique (R_int_ = 0.1153) which were used in all calculations. The final *wR_2_* was 0.3316 (all data) and *R_1_* was 0.1147 (I ≥ 2 *s*(I)).

### 4.7. Bioassays

Test compounds were dissolved in DMSO to provide 10 mg/mL stock solutions. An aliquot of each stock solution was transferred to the first lane of rows B to G in a 96-well microtiter plate and 2-fold serially diluted across the 12 lanes of the plate to provide a 2048-fold concentration gradient. Bioassay medium was added to an aliquot of each test solution to provide a 100-fold dilution into the final bioassay, thus yielding a test range of 100 to 0.05 μg/mL in 1% DMSO. Row A was used as the negative control (no inhibition), and Row H was used as the positive control (uninoculated).

CyTOX is an indicative bioassay platform for discovery of antitumor actives. NS-1 (ATCC TIB-18) mouse myeloma cells and NFF (ATCC PCS-201) human neonatal foreskin fibroblast cells were each inoculated in 96-well microtiter plates (190 μL) at 50,000 cells/mL in DMEM (Dulbecco’s modified Eagle’s medium + 10% fetal bovine serum (FBS) + 1% penicillin/streptomycin (10,000 U/mL/10,000 µg/mL, Life Technologies Cat. No. 15140122), together with resazurin (250 µg/mL; 10 µL) and incubated in a 37 °C (5% CO_2_) incubator. The plates were incubated for 96 h during which time the positive control wells change colour from a blue to pink colour. MIC end points were determined visually.

ProTOX is a generic bioassay platform for antibiotic discovery. *Bacillus subtilis* (ATCC 6633) and *Staphylococcus aureus* (ATCC 25923) were used as indicative species for antibacterial activity. A bacterial suspension (50 mL in a 250 mL flask) was prepared in nutrient broth by cultivation for 24 h at 100–250 rpm, 28 °C. The suspension was diluted to an absorbance of 0.01 absorbance unit per mL, and 10 μL aliquots were added to the wells of a 96-well microtiter plate, which contained the test compounds dispersed in nutrient agar (Amyl) with resazurin (12.5 μg/mL). The plates were incubated at 28 °C for 48 h, during which time the negative control wells change from a blue to light pink color. MIC end points were determined visually.

EuTOX is a generic bioassay platform for antifungal discovery. The yeast *Candida albicans* (ATCC 10231) and *Saccharomyces cerevisiae* (ATCC 9763) wereused as indicative species for antifungal activity. A yeast suspension (50 mL in a 250 mL flask) was prepared in 1% malt extract broth by cultivation for 24 h at 250 rpm, 24 °C. The suspension was diluted to an absorbance of 0.005 and 0.03 absorbance units per mL for *C. albicans* and *S. cerevisiae*, respectively. Aliquots (20 µL and 30 µL) of *C. albicans* and *S. cerevisiae*, respectively, were applied to the wells of a 96-well microtiter plate, which contained the test compounds dispersed in malt extract agar containing bromocresol green (50 μg/mL). The plates were incubated at 24 °C for 48 h, during which time the negative control wells change from a blue to yellow color. MIC end points were determined visually.

GiTOX is a bioassay focused on the discovery of inhibitors of the parasite, *G. duodenalis*. In the present bioassay *G. duodenalis* (strain WB-1B) was inoculated in 96-well microtitre plates (200 µL) at 4 × 105 cells/mL in Giardia medium (0.2% tryptone, Oxoid; 0.1% yeast extract, Difco; 0.5% glucose; 0.106% L-arginine; 0.1% L-cysteine; 0.2% NaCl; 0.1% K_2_HPO_4_; 0.06% KH_2_PO_4_; 0.02% ascorbic acid; 0.0023% ferric ammonium citrate; 0.01% Bile (Sigma, Burlington, MA, USA); 1% penicillin/streptomycin (10,000 U/mL/10,000 µg/mL, Life Technologies Cat. No. 15140122), 10% newborn calf serum (NBCS), Life Technologies, Waltham, MA, USA). The plates were incubated in anaerobic jars (Oxoid AG25) containing an Anaerogen satchel (Oxoid AN25) in a 37 °C (5% CO_2_) incubator. At 96 h, *G. duodenalis* proliferation was counted and % inhibition graphed to determine the MIC values.

TriTOX is a bioassay focused on the discovery of inhibitors of the animal protozoan pathogen *Tritrichomonas fetus* (strain KV-1). *T. fetus* were inoculated in 96-well microtiter plates (200 μL) at 4 × 10^4^ cells/mL in *T. fetus* medium (0.2% tryptone, Oxoid; 0.1% yeast extract, Difco; 0.25% glucose; 0.1% l-cysteine; 0.1% K_2_HPO_4_; 0.1% KH_2_PO_4_; 0.1% ascorbic acid; 0.01% FeSO_4_·7H_2_O; 1% penicillin/streptomycin (10 mL/L), 10% new born calf serum, Life Technologies, Waltham, MA, USA). The plates were incubated in anaerobic jars (Oxoid AG25) containing an Anaerogen satchel (Oxoid AN25) in a 37 °C (5% CO_2_) incubator. At 48 h and 72 h, MIC end points were determined visually and absorbance was measured using Spectromax plate reader (Molecular Devices, Sunnyvale, CA, USA) at 570 nm [21].

## Figures and Tables

**Figure 1 marinedrugs-21-00041-f001:**
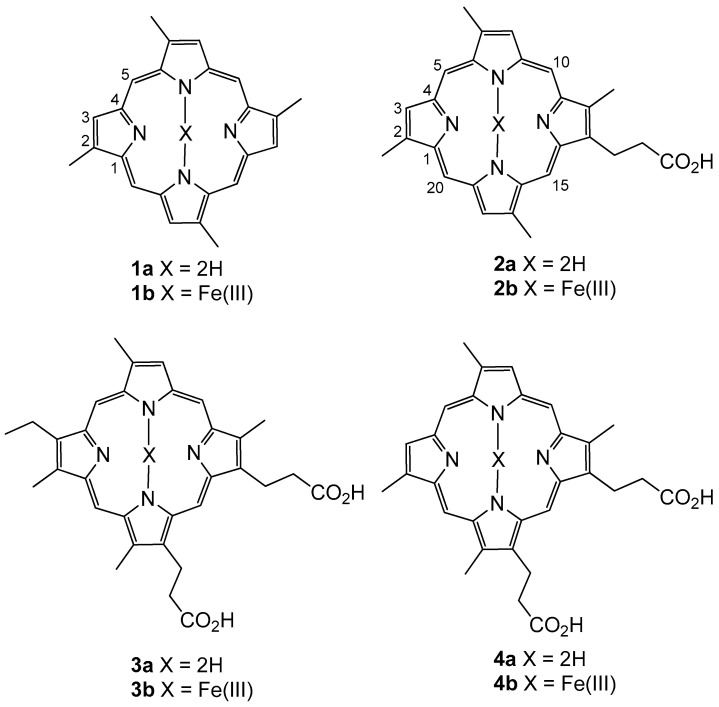
Structures **1a/b** to **4a/b**.

**Figure 2 marinedrugs-21-00041-f002:**
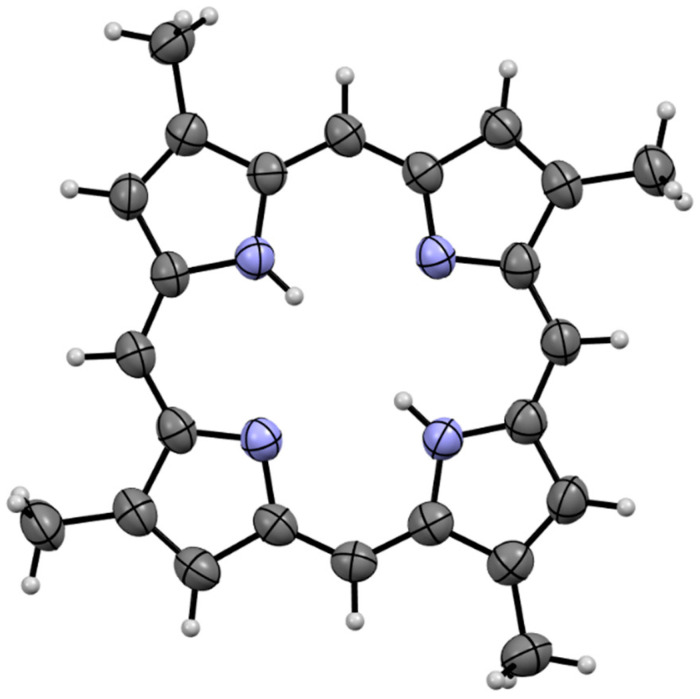
Single crystal X-ray structure of **1a**.

**Figure 3 marinedrugs-21-00041-f003:**
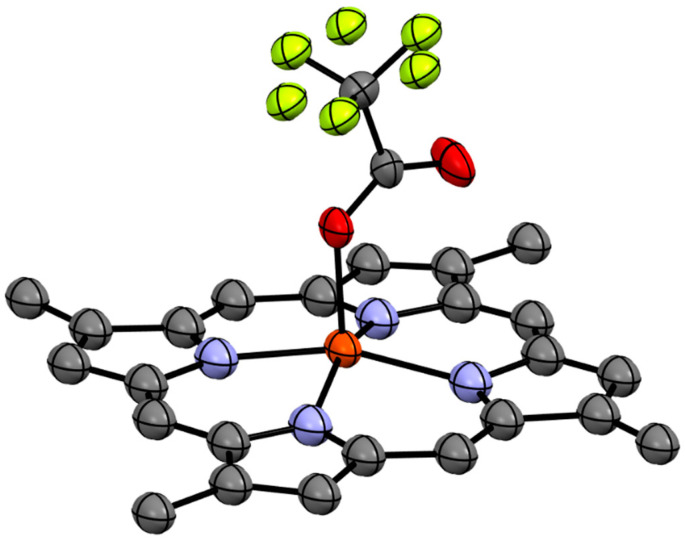
Single crystal X-ray structure of **1b** TFA salt. Hydrogen atoms have been omitted for clarity.

**Figure 4 marinedrugs-21-00041-f004:**
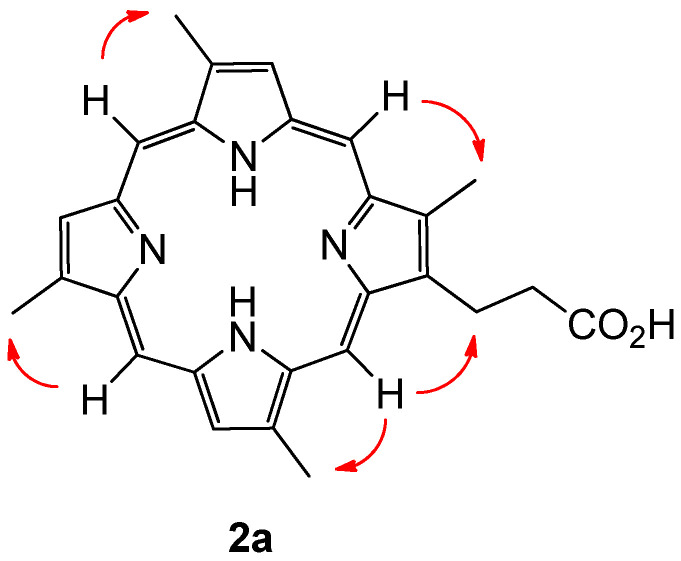
Isabellin B (**2a**) with key NOE correlations indicated in red.

**Table 1 marinedrugs-21-00041-t001:** Porphyrin-hemin mass correlation table for compounds **1a/b**–**4a/b**.

Compound Pair	[M + 2H + H]^+^	Molecular Formula (a)	[M + Fe + ACN]^+^	[M + Fe]^+^	Molecular Formula (b)
**1a/b**	367.1924	C_24_H_22_N_4_	461.1302	420.1032	C_24_H_20_N_4_^56^Fe^+^
**2a/b**	439.2133	C_27_H_26_N_4_O_2_	533.1522	492.1248	C_27_H_24_N_4_O_2_^56^Fe^+^
**3a/b**	539.2653	C_32_H_34_N_4_O_4_	633.2038	592.1772	C_32_H_32_N_4_O_4_^56^Fe^+^
**4a/b**	511.2345	C_30_H_30_N_4_O_4_	605.1718	564.1474	C_30_H_28_N_4_O_4_^56^Fe^+^

**Table 2 marinedrugs-21-00041-t002:** Bioactivity profile of isabellins A and B (**1a** and **2a**), isabellihemin A (**1b**), corallistin D (**3a**) and deuteroporphyrin IX (**4a**) against select Bacteria and Fungi, MIC reported in μg/mL. ^a^.

Compound	*Bs 24 h*	*Bs 48 h*	*Sa 24 h*	*Sa 48 h*	*Ca 24 h*	*Ca 48 h*	*Sc 24 h*	*Sc 48 h*
**1a**	25	>100	25	>100	>200	>200	>200	>200
**1b**	>100	>100	>100	>100	>200	>200	>200	>200
**2a**	>100	>100	>100	>100	>200	>200	>200	>200
**3a**	>100	>100	>100	>100	>200	>200	>200	>200
**4a**	100	100	50	100	>200	>200	>200	>200
Control ^b^	1.6	6.3	3.1	12.5	0.8	>200	1.6	3.1

^a^*Bs* = *Bacillus subtilis* (ATCC 6633); *Sa* = *Staphylococcus aureus* (ATCC 25923); *Ca* = *Candida albicans* (ATCC 10231); *Saccharomyces cerevisiae* (ATCC 9763); ^b^ Controls: *Bs, Sa* = tetracycline; *Ca, Sc* = blasticidin S HCl.

**Table 3 marinedrugs-21-00041-t003:** Bioactivity profile of isabellins A and B (**1a** and **2a**), isabellihemin A (**1b**), corallistin D (**3a**) and deuteroporphyrin IX (**4a**) against select cell lines and protozoal parasites, MIC reported in μg/mL ^a^.

Compound	NS-1 *72 h*	NS-1 *96 h*	NFF *72 h*	NFF *96 h*	*Tf 48 h*	*Tf 72 h*	*Gi 96 h*
**1a**	0.4	0.8	1.6	1.6	>100	>100	6.3
**1b**	>100	>100	>100	>100	>100	>100	100
**2a**	>100	>100	>100	>100	>100	>100	>100
**3a**	50	100	>100	>100	>100	>100	100
**4a**	>100	>100	>100	>100	>100	>100	100
Control ^b^	0.6	0.6	0.6	0.6	0.2	0.2	0.3

^a^ NS-1 = Murine myeloma NS-1 (ATCC TIB-18); NFF = Neonatal foreskin fibroblast (ATCC PCS-201); *Tf* = *Tritrichomonas foetus* KV-1; *Gi* = *Giardia duodenalis* WB-1B. ^b^ Controls: NS-1, NFF = sparsomycin; Tf, Gi = metronidazole.

## Data Availability

Data available from zenodo: DOI:10.5281/zenodo.7425830. Link: https://zenodo.org/record/7425830#.Y5aOtJ5BxPY.

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
