# Peer review of "Type-I Hemins and Free Porphyrins from a Western Australian Sponge Isabela sp."

_marinedrugs, 2023, doi:10.3390/md21010041_

Round 1

Reviewer 1 Report

This is an excellently presented manuscript that articulates the purification and structure ID of molecules that are difficult to isolate and characterize. The information presented for the structure determination is easy to follow and well done.

The only issue I have with the manuscript is the use of the term selective toxin. The cytotoxicity is only tested in two cells and the level of activity is not that different. I would expect to see a 5-10 fold difference in cytotoxicity to say selective.

Author Response

>This is an excellently presented manuscript that articulates the purification and structure ID of molecules that are difficult to isolate and characterize. The information presented for the structure determination is easy to follow and well done.

The authors thank the reviewer for the positive review.

>The only issue I have with the manuscript is the use of the term selective toxin. The cytotoxicity is only tested in two cells and the level of activity is not that different. I would expect to see a 5-10 fold difference in cytotoxicity to say selective.

Here, the selectivity intended in the text was a reflection of selectivity for mammalian cell lines over other eukaryotic organisms (Tritrichomonas foetus, Giardia duodenalis, Candida albicans  and Saccharomyces cerevisiae) as well as the Gram-positive bacterial pathogens tested (Bacillus subtilis and Staphylococcus aureus). The manuscript has been amended to reflect this observation.

Reviewer 2 Report

1. The name of the taxonomist who identified the sponge material should be acknowledged in the manuscript.

2. The code of the voucher sample of the sponge and its location should be mentioned along with the description of the sponge.

3. An underwater photograph of the sponge either in the manuscript or supplementary data  would be helpful

Author Response

We thank the reviewer for the timely review.

>The name of the taxonomist who identified the sponge material should be acknowledged in the manuscript.

The taxonomist who identified the sponge is a co-author on the manuscript (Jane fromont)

>The code of the voucher sample of the sponge and its location should be mentioned along with the description of the sponge.

The section "animal material" describes the voucher code and location of type specimen as follows:

"A specimen of Isabela sp. (WAM Z35787) was collected at 97 metres depth on hard substrate off Zuytdorp, WA (27°03`06"S, 113°06`03"E) by Sherman sled on 5/12/2005 aboard the CSIRO research vessel Southern Surveyor, and was stored frozen at -18 °C at the Western Australian Museum"

>An underwater photograph of the sponge either in the manuscript or supplementary data would be helpful

The sponge was collected from a depth of 97 m, by Sherman sled. as such, acquisition of an underwater photograph of the species is unfeasible.

Reviewer 3 Report

This manuscript described two novel free porphyrins isolated from a marine sponge Isabela sp. The new compounds were identified by high-resolution mass spectrometry, UV-Vis 18 spectroscopy and extensive NOESY NMR spectroscopic experiments. Compound 1a showed cytotoxic activities against NS-1 and NFF cell lines. This work is interesting and I think the paper would be worth for publication in the Journal after the following minor concerns should be addressed:

1. Page 4, line 111, the 'd' in 'DMSO-d6' should be italic.

2. Page 4, line 117, what are the main differences between type-I and type-III derived porphyrin macrocycles? The authors should introduce it here.

3. Page 5, should the cytotoxic activities be evaluated by IC50 instead of MIC?

Author Response

We thank the reviewer for the timely review

>Page 4, line 111, the 'd' in 'DMSO-d6' should be italic.

The line has been ammended.

>Page 4, line 117, what are the main differences between type-I and type-III derived porphyrin macrocycles? The authors should introduce it here.

The text has been ammended as follows:

"Given the capacity of the organism in question to produce porphyrins with both type-I (derived from a symmetrical APAPAPAP bearing uroporphirinogen precursor, with A = Acetyl and P = Propionyl) and type-III (derived from an asymmetrical APAPAPPA bearing uroporphirinogen precursor, featuring D-ring inversion) topology, ..." 

>Page 5, should the cytotoxic activities be evaluated by IC50 instead of MIC?

Cytotoxicity has been quoted as MIC values to allow for easy comparison of biological activity across organisms and cell lines.